# Perceptions of Youth Substance Users on Substance Use Relapse Prevention: A Qualitative Study in Lobatse, Botswana

**DOI:** 10.3390/ijerph23010062

**Published:** 2025-12-31

**Authors:** Wada Gaolaolwe, Miriam Mmamphamo Moagi, Gaotswake Patience Kovane, Leepile Alfred Sehularo

**Affiliations:** 1NuMIQ Research Focus Area, Faculty of Health Sciences, North-West University, Mahikeng Private Bag X2046, South Africa; patience.kovane@nwu.ac.za; 2School of Nursing, Faculty of Health Sciences, University of Botswana, Gaborone Private Bag 00712, Botswana; 3Nursing Department, University of Limpopo, Sovenga Private Bag X1106, South Africa; miriam.moagi@ul.ac.za; 4Lifestyle Diseases Research Focus Area, Faculty of Health Sciences, North-West University, Mahikeng Private Bag X2046, South Africa; leepile.sehularo@nwu.ac.za

**Keywords:** substance use, relapse, youth, prevention, addiction, recovery

## Abstract

**Background**: Substance use relapse is a significant obstacle that hinders the success of addiction treatment and recovery for youths struggling with substance use challenges. The economic and social impacts of substance use and relapse in young people are a cause of concern worldwide; Botswana is not an exception. **Objective**: The study aimed to explore and describe the perceptions of youth substance users regarding the prevention of relapse in Lobatse, Botswana. **Methods**: A qualitative study was conducted, following an exploratory, descriptive, and contextual research design. In total, 15 participants were selected using a purposive sampling technique. Data were collected using semi-structured questions, and thematic analysis was used to analyse the data. **Results**: Data analysis yielded two themes and subthemes. The first theme was on the perceptions of youth substance users on the causes of substance use relapse. The second theme was on the perceptions of youth substance users on the prevention of substance use relapse. **Conclusion**: The study showed that substance use relapse can be caused by psychological challenges, social problems, societal issues, and healthcare barriers. Our study suggests that to abate relapses to substances, interventions should encompass individual, social, and community dimensions of a substance user’s life. Furthermore, there should be healthcare interventions geared towards relapse prevention.

## 1. Introduction

Substance use relapse (SUR) is a major challenge that militates against addiction treatment and recovery successes for substance users. The inability to sustain abstinence, particularly for longer periods, has worse outcomes, such as lower rates of retention in treatment and health and social impairment. Worldwide, relapse into substance use following treatment is common and afflicts the youth most [1,2,3]. The prevalence of substance use among the youth exceeds that of other age groups, and it is often where it begins. Volkow et al. [4] found that the lifetime prevalence of substance use among the youth was 79.7% for alcohol, 51.5% for cannabis, and 55.0% for tobacco. In the United States (US), Jones [5] reported 29.2% alcohol use, 13.7% binge drinking, and 21.7% use of marijuana among the youth. However, substance use is more common in Africa, and the highest rate of heavy consumption of alcohol globally is in sub-Saharan Africa [6]. A study conducted in Tanzania on alcohol use confirmed a significant drinking problem among the youths (15–24 years), with 71% of males reporting heavy episodic drinking as compared to 27% amongst females [7]. In Botswana, Riva et al. [8] reported 42.1% of alcohol use among the youths, more than twice the results of other studies in Botswana, and 16.7% use of illicit drugs.

Initiating substance use at an early age has been shown to increase the likelihood of future relapses significantly. About 65–85% of young persons with substance use disorders (SUD) experience a relapse 12 months after starting treatment [3,9]. In Botswana, SUR research among the youth has not been actively conducted. However, anecdotal information indicates that many YSUs relapse into substance use after completing a recovery process. This offered a critical chance to address a serious problem that has not yet received enough research attention.

Moreover, it is worrying for Botswana that certain sub-Saharan African countries have a SUR rate as high as 59.9% to 75% [2,10,11]. Albeit the urgent need to prevent the use of substances and SUR, the government of Botswana has placed considerable efforts on the former, with programmes such as “ke sharp”, “fokotsa dino” (I am fine; reduce drinking), but little attention has been given to SUR prevention. Thus, relapses are rampant, putting a strain on families, society, and healthcare systems.

As stated by Degenhardt et al. [12], the economic and societal ramifications of substance use and relapse to use in young people are substantial.

During the early and middle years of development, young people undergo significant changes in various aspects of their lives, including biological, cognitive, social, and emotional factors that can influence the formation of their self-identity. At this stage, young people may begin using substances for a variety of reasons, such as to boost their self-esteem, for hedonic purposes, or to cope with psychosocial issues [11,13]. Unfortunately, the start of substance use at an early age is highly linked to addiction and persistent relapses of use. When a person experiences SUR, it is a setback in which they fail to successfully alter their substance use behaviours and resume using drugs or alcohol after a time of sobriety [14]. In contrast to the prevalent perspective, which conceptualises “relapse” as either “abstained” or “relapsed,” with abstinence being the most crucial component of rehabilitation, this term is all-inclusive [14].

Throughout the journey of youth recovery, SUR can exert a substantial influence on recovery progress, often manifesting in recurring patterns of abstinence and relapse [14,15]. Yet, it is crucial to acknowledge that a relapse is not solely a lack of determination on the part of the youth substance user (YSU) [16]. SUR is intricate and numerous factors, including inadequate familial support, peer pressure, failure to resist cravings, low socioeconomic status, and lack of robust relapse prevention strategies, are linked to substance use relapse among the youth [14,17,18]. Studies conducted by Jenkins et al. [13] and Swanepoel et al. [11] concur that despite the variety of causes for substance use relapse among YSUs, the recurrences are predominantly for hedonistic purposes.

In Botswana, the negative urbanisation consequences, such as substance use, delinquent groups, particularly among young people, have plagued the Southeast area, which includes Lobatse, as a result of Gaborone’s rapid economic expansion as the country’s capital city and a high economic zone. This rapid growth is accompanied by the proliferation of drugs, a factor that contributes to high relapse rates. In response to this problem, there are some non-governmental organisations (NGOs) whose aim is to fight the drug use epidemic, but they lack intensive rehabilitation programmes, such as inpatient services. There is also a lack of community-based centres, youth-friendly halfway homes and robust outpatient care facilities to promote the uptake of SUD services by the YSUs. The government hospitals provide general psychiatric services, and the country lacks facilities specialised in substance use rehabilitation. All these problems result in a gap in the provision of both health and social services for substance users in the country.

In this study, understanding the perceptions of the YSUs about important substance use relapse prevention behaviours and essential recovery needs can provide insight for the treatment community working to address substance use issues.

## 2. Methods

### 2.1. Research Design

A qualitative, exploratory, and descriptive research design was used to explore and describe the perceptions of youth substance users on the prevention of substance use relapse in Lobatse, Botswana.

### 2.2. Study Setting

This study was conducted at an outpatient SUD clinic of a referral psychiatric hospital in Lobatse, a town in Botswana’s Southeast district. The hospital offers both outpatient and inpatient psychiatric mental health services. Medical reviews of patients with substance use disorders are conducted at a substance-use disorders clinic in the Outpatient Department (OPD). The youth who were deemed to have experienced SUR are those who experienced a setback in their attempt to change substance use behaviours and started to use substances after a period of abstinence. The selected hospital was identified for the study because it is the country’s only government-owned referral psychiatric hospital, a tertiary institution that provides psychiatric mental health rehabilitation services and has high value for research.

### 2.3. Study Population

The study was conducted on YSUs who met the eligibility criteria and came to the OPD for medical reviews at the substance use disorders clinic. The participants were required to have a history of attempting to quit substances in the past but experienced a relapse.

### 2.4. Sampling

The sample was selected homogenously from participants who shared a set of characteristics required for the study. A purposive sampling technique was used to select YSUs who met the eligibility criteria for the study. This sampling technique enabled the researcher to draw a sample of YSUs who experienced SUR. The sample size for the YSUs was 15 and was determined by data saturation, when no new themes emerged in the data, and redundancy was realised.

#### 2.4.1. Inclusion Criteria

Participants were between 18 and 24 years, in line with the definition of the United Nations (UN), which places the youth category between 18 and 24 years. This is generally the time when education is compulsory to obtain the first job. Only participants who could speak English or Setswana were included in the study. The participants had a history of attempting to quit substances at some point and experienced a relapse.

#### 2.4.2. Exclusion Criteria

The youth who experienced a relapse within the last three months and those who were admitted for a substance use problem but never attempted to quit drugs were excluded from the study. Also, YSUs who met the eligibility criteria but were unwilling to be audio recorded for data collection or to sign an informed consent form to participate in the study were excluded.

### 2.5. Recruitment Process

The hospital superintendent in charge of the selected hospital was the gatekeeper of the study. Firstly, the gatekeeper was furnished with a letter from the Ministry of Health, Botswana, that permitted the researcher to conduct the study at the institution, followed by a detailed explanation of the study. Then announcements on the study were made to clients at the OPD for recruitment. Furthermore, posters were displayed in hospital notice boards with an explanation of the study to sensitise prospective participants. The posters contained information that explained the purpose of the study, the risks and benefits of participating in the study, and the contact details of the independent person whose purpose was to negotiate consent.

Thus, the independent person was not directly involved in the study and was used to negotiate consent with the participants. The researcher comprehensively explained the study to the independent person, including his roles. The independent person had a master’s degree qualification, with experience in conducting both data collection and analysis.

During consent negotiation, the participants were informed of their rights to refrain from participation and to withdraw or terminate participation at any point of the study with no repercussions. Following verbal consent, eligible participants sealed their willingness to participate with a signed written consent to the independent person before data collection by the researcher.

### 2.6. Data Collection Method

Semi-structured individual face-to-face interviews were used to collect data from YSUs regarding their perceptions of SUR prevention. Data was gathered between April and July of 2024. One of the hospital’s OPD consulting rooms served as the interview setting. The interviews were conducted at times suitable for the participants. The participants were provided with refreshments during the interview. To reduce the youth participants’ risk of being stigmatised, the consultation room was away from the regular traffic of people in the OPD. There was a “do not disturb” notice on the door for controlled access to the interview room. The individual interviews were conducted in Setswana or English, depending on the participants’ preferences. They were audio recorded, using a semi-structured interview guide that was first created in English and then translated into Setswana. Fifteen interviews were conducted, of which fourteen (14) were conducted in Setswana, while one interview was conducted in English. Every interview lasted between forty-five (45) and an hour. During the interviews, interpersonal communication strategies, including exploration, asking for clarification, probing, and reflecting, were employed. The following interview questions were asked:


*“As a youth, what are your experiences of substance use relapse among the youth?”*



*“What do you think can be done to prevent substance use relapse among the youth?”*


### 2.7. Ethical Considerations

The study commenced after the researcher received ethical approval from the Institutional Review Board of North-Western University (Ethics number: NWU-00174-23-A1). Permission to conduct the study was also obtained from the Ministry of Health, Botswana (Ref: HPRD:6/14/1) and the referral psychiatric hospital in Lobatse [Ref: 4/2/2 III (65)]. After obtaining informed written consent from the participants, data were collected. To ensure anonymity, no identifying information was used in the data, and the data file was password-protected, with access restricted to the lead investigator.

### 2.8. Data Analysis

Data was analysed using the six-phase approach of thematic data analysis by Braun and Clarke [19]. The interviews that were conducted in the Setswana language were transcribed verbatim and translated into English for analysis. The raw data and the study protocol were sent to the co-coder for an independent analysis of the data by both the researcher and the co-coder, who was an experienced qualitative researcher. Themes were identified using abbreviations and symbols for word classification. A meeting was then arranged between the researcher and the co-coder to reach a consensus on the themes and categories.

During the first phase of thematic analysis, which involves familiarising oneself with the data, the researcher and co-coder immersed themselves in the data set through reading and rereading it, and at the same time searching for patterns of meaning [19]. It is through the exercise of reading and rereading transcripts that the duo gained a deeper understanding of the data and key issues that concerned the phenomenon studied.

The second phase was generating initial codes. At this stage, the data was systematically analysed using codes and was organised into meaningful groups. It is in this phase that all the data extracts were coded, and those that bore the same code were assembled. The labels were assigned to the data that were deemed potentially relevant to the research question.

The third phase involved searching for themes. The focus shifted from codes to themes in this phase. As posited by Braun and Clarke [19], codes provide some pattern of responses in a data set while themes capture what is regarded as important in a given data set, e.g., what answers a research question. Thus, themes were generated by reviewing the coded data and identifying overlaps, similarities, and differences between codes. The fourth phase involved reviewing potential themes and is concerned with quality checks. Thus, reviewing potential themes was a recursive process, in which emerging themes were appraised against the coded data, including the data set in its entirety.

The fifth phase was defining and naming the themes. This is the phase in which the researcher reviewed the names of themes and specified their essence. Finalisation of themes took place in this phase after their renaming had led to satisfactory results. As Braun and Clarke [19] suggest, the researcher developed the final informative and concise themes. The sixth last phase was producing the report. At the end, the researcher presented a story with themes that were coherent and connected to each other. Producing a report involved choosing, ordering, and chronicling the study findings that yielded an account of the data [19].

### 2.9. Trustworthiness

The rigour of data collection and data analysis was ensured by meticulously following Lincoln and Guba’s framework [20], in which the credibility of the data was guaranteed through listening to the interview audios repeatedly. Credibility of the study was realised through the reading and rereading of transcripts for accurate description and interpretation of the perceptions that the YSUs had concerning the prevention of SUR. A thorough review of the interview recordings and transcripts was carried out to enhance credibility. For dependability, which is the stability of data over time and conditions, consensus on themes was achieved through a stepwise replication strategy by involving a co-coder in the independent analysis of data [20]. The results were later compared for consensus on themes. Confirmability was also ensured in the study, and it entails the extent to which the findings of a study can be certified by other researchers [20]. As such, an audit procedure was implemented throughout the study to validate the transcripts, guaranteeing confirmability. Trustworthiness was also ensured through transferability, which is the extent to which the study results can be extrapolated to other settings [20]. To address transferability, a dense description of the study population, including their demographics, was provided, and the findings were presented in detail within the study context.

## 3. Findings

### 3.1. Demographic Description

A total of fifteen youths (N = 15) diagnosed with SUD took part in the study. Five participants were females, while the rest were males, constituting the largest group in the sample (n = 10). All participants were black, aged 18 to 24 years, with a mean age of 22 years (std. dev. 2). Of the 15 participants, one had primary education, seven had secondary education, and the remaining seven had tertiary education (Table 1).

### 3.2. Organisation of the Themes

Two themes and subthemes for the themes on the perceptions of YSUs regarding substance use relapse prevention were extracted from the data (Table 2).

#### 3.2.1. Theme 1: Perceptions of Youth Substance Users on the Causes of Substance Use Relapse

The participants articulated their perspectives on the causes of SUR. They identified psychological, social, and societal causes as well as healthcare barriers as potential reasons for SUR. The subsequent sections outline the participants’ insights into the causes of SUR, accompanied by relevant quotes.

##### Subtheme 1.1: Psychological Causes of Substance Use Relapse

The study identified that youth substance users perceive their relapse to substances to be emanating from psychological causes, as substantiated by the following quotes:


*“Challenges we face in life cause all these drawbacks.”*
[P# 1]


*“When I experience physical and psychological problems, I go back to drugs to give me a better feeling, and life goes on.”*
[P# 5]


*“The mind is very powerful. Whatever we think about determines the outcomes of our actions. When I think a lot about drugs, it causes or increases the cravings and the likelihood of relapsing to use becomes high.”*
[P# 8]


*“That is why we relapse after attempting to quit. We try to stop drugs when the problem that is the root cause of one’s substance use is not resolved. If for some reason the problems I have resurface again in my mind and makes me feel low, I would go back to drugs because I know I will get an instant fix. And that is very easy if I am in an environment of people who are using.”*
[P# 10]

##### Subtheme 1.2: Social Causes of Substance Use Relapse

The findings of this study show that the youth perceived their relapse to substances to be caused by the negative social circumstances they found themselves in. Negative influence from friends, family rejection, and lack of family involvement were perceived as responsible for the relapses, as exemplified in the following quotes:


*“You can also relapse back to substances because of friends…Friends will tell you that drugs or alcohol are not an employment or a job that you talk about quitting.”*
[P# 1]


*“Lack of support at home. Our parents give up on us early. They don’t even have an interest in wanting to know why you started using drugs. They just blame and label you for using drugs and don’t want to know your issues why you are on drugs.”*
[P# 6]


*“You feel you belong there, because there nobody judges you, nobody tells you that you are a drug addict. There you are at the same level, and a person may also borrow something from you like a cigarette lighter and that gives you feel good. You see… there you feel that this is where I belong… nobody judges me here… we are all at the same level.”*
[P# 6]


*“At times you find that if I have a conflict with the family, but I get admitted for substance use, I get counselled alone and the family is left out. We have to be assisted as a family. If I am counselled while the actual problem I left it home is a challenge.”*
[P# 10]


*“You know sometimes you would make a mistake and you are told, ‘it’s you who is useless and can’t even do anything to assist yourself’. Some of these carelessly tossed words that are degrading makes you lose hope and you tell yourself, ‘I’m am already useless, why bother… why not use’.”*
[P# 13]

##### Subtheme 1.3: Healthcare Barriers to Prevent Substance Use Relapse

The findings of this study show that the lack of patient follow-ups to monitor their progress post-discharge is responsible for their relapses. Due to this gap, there is a lack of information on how families and communities contribute to relapses to substance use.


*“So, the gap is that health workers are not visible in the communities to address issues that affect the community… they operate at their facilities only and this is a gap, and we end up lacking crucial health information.”*
[P# 3]


*“Lack of follow-ups can lead to substance use relapses. If a youth is admitted for substance youth rehabilitation and gets discharged, it should not be concluded that the substance user benefitted from the rehabilitation and will be abstinent.”*
[P# 4]

#### 3.2.2. Theme 2: Perceptions of Youth Substance Users on the Prevention of Substance Use Relapse

The findings of this study show that effective youth substance use relapse prevention requires preventive interventions at individual, social, and community levels. Also, the results showed important healthcare interventions that can be used to prevent substance use relapses. The subthemes that follow and related quotes reflect the youths’ perceptions of SUR prevention.

##### Subtheme 2.1. Individual Interventions to Prevent Substance Use Relapse

The YSUs discussed self-initiated interventions that can prevent substance use relapse at an individual level. These interventions include, inter alia, identifying and engaging in pleasurable activities to distract from cravings.


*“Cravings are difficult to deal with. And cravings are common in people who use drugs. But to deal with cravings, a substance user must identify some activities to always engage in as a way of replacing the drug by keeping busy and distracting oneself from thinking about the substance.”*
[P# 1]


*“You wake up in the morning knowing what you are supposed to do gives you purpose in life. You think about productivity at work and keeps your mind thinking about work activities distracts a person from thinking about drugs.”*
[P# 11]


*“Motivation… It must also come from within an individual to want to quit. People can’t force a person to quit, but it must be a decision and commitment from a substance user.”*
[P# 13]

##### Subtheme 2.2. Social Interventions to Prevent Substance Use Relapse (Social Support)

The participants shared their perceptions on the social interventions that can be beneficial in preventing SUR, with the main emphasis being the need for social support, as attested in the quotes below.


*“Look, there is a phrase that says, “If one of your eyes is giving you trouble, remove it and remain with one and if your eyes are a problem, remove all of them”. If all your friends are not good for you, just remove them all from your life.”*
[P# 1]


*“As for money, you would need someone who will assist you in managing your money. The person should be trusted preferably a close relative who shall hold your money and together with you ensure that you follow the budget you agreed upon. If there is any change from the drawn budget, it should be used on other planned things. With money, it is important to engage someone you trust who is a family member or relative to assist you in managing and using your finances appropriately.”*
[P# 6]


*“Social support from both friends and family is important. Before I started rehabilitation, my family was no longer trusting me. But since I started rehabilitation, they can see that I intend to change, and I am starting afresh. I nearly lost direction, but I am correcting my life to be in the right direction.”*
[P# 3]

##### Subtheme 2.3. Community Interventions to Prevent Substance Use Relapse (Community Awareness and Education)

The participants discussed community interventions to prevent SUR. Public education and awareness campaigns were highlighted, as shown in the quotes.


*“Public health campaigns and education on the dangers of using substances can prevent relapses. Public education on the dangers of using substances is seriously lacking. Public health education must be intense because there are a lot of drugs now in our communities.”*
[P# 1]


*“They must be given education to understand that we are their children, we are also people just like them. What happens is that we are criticized which makes us go back to drugs and our lives become a vicious circle.”*
[P# 5]


*“The same applies to mental health… if people can be well informed about substance use, the stigma can be reduced. Like it happened with the stigma for HIV/AIDS.”*
[P# 8]

##### Subtheme 2.4. Healthcare Interventions to Prevent Substance Use Relapse

The participants shared their views on healthcare interventions to prevent SUR, as shown in the following quotes:


*“We need a facility that is independent of the psychiatric hospital and people will not fear to ask for help as is the case now because they say this is a place for those who are mad. There is a need for facilities that are specifically for those with substance use problems.”*
[P# 1]


*“Mm, the information on the biology of the brain was interesting and eye-opening. It really opened my eyes. Eye-opening in the sense that you cannot want to continue using drugs with all this information on how the drugs affect your body.”*
[P# 4]


*“It’s crucial to provide ongoing health education, follow-ups and counselling to ensure that individuals maintain abstinence from drugs after completing rehabilitation.”*
[P# 7]


*“Occupational therapy should be continued even after discharge. Activities like sewing, carpentry, and leather works should be provided to those in the recovery process and are outpatients. These activities can keep the youth busy and prevent use of drugs.”*
[P# 11]

## 4. Discussion

The findings of this study revealed that the causes of substance use relapse are multifaceted and occur at different levels, such as psychological, social, and societal. Additionally, the barriers in healthcare were also found to cause substance use relapse. At a psychological level, participants in the current study reused substances after quitting to cope with their stressors, a phenomenon that literature refers to as stress coping [21]. With stress coping, the use of substances is viewed as a coping response to life stressors that increase positive affect or alleviate negative affect [21]. A study by Yang et al. [17] yielded similar findings in which people who quit substances relapsed to use when confronted with some stressful life events as a means of coping.

At the social level, the results of this study showed SUR to be emerging from different yet interrelated social circumstances. The youth perceived their relapses as emanating from social circumstances such as lack of family support, family conflicts, and peer pressure. YSUs who experienced a lack of family support and were embroiled in family conflicts found themselves without a safety net, which led them to seek solace in substances and peers who used drugs as a form of escape from familial tensions and to have a sense of belonging. This finding is corroborated by those of a study by Atadokht et al. [22] in which negative family members’ expressions of emotions towards persons with substance use problems resulted in their return to the old form and relapse to substances. The pressures from negative criticism and lack of family support were reported to put a heavy strain on individuals with SUDs, leading to a regression to the past negative life and relapse to substance use to cope [22]. However, in the present study, peer pressure also presented another layer of SUR where YSUs succumbed to the influence of their social circles and felt compelled to engage in substance use to gain acceptance and fit in the group. Similarly, participants in a study by Uwera et al. [23] reported relapse to substances by youth due to peer influence.

There is evidence from this study that the lack of follow-ups by the healthcare providers after a patient has been discharged from inpatient rehabilitation contributes to SUR. The participants in this study reported a lack of ongoing support and follow-up care that would facilitate assessment and interventions to prevent SUR. Previous studies have also shown that patients with substance use problems who are at high risk of relapse need follow-up care during their recovery trajectory [24,25]. The findings of this study also revealed the perceptions that the YSUs hold on how SUR can be prevented.

Beyond evidence-based clinical interventions, this study found that individual interventions are necessary to prevent SUR among the YSUs. Appiah et al. [10] also found that self-initiated interventions, such as taking a walk, etc., appeared effective in SUR prevention among participants who initiated and utilised them. In the current study, individual interventions reported to prevent SUR include engagement in pleasurable activities to distract cravings, and motivation to quit on the side of the YSU. These findings lend support to extant literature. Welsh and Hadland [26] suggest that to build recovery capital and prevent SUR, the YSUs need to identify activities that they find fun or can increase pleasure in their recovery. Also, Furzer et al. [9] underscore the importance of youths’ engagement in the physical activities they enjoy in the prevention of SUR. In addition to engaging in pleasurable activities, internal motivation to quit was reported to be important in SUR prevention in the current study, contrary to the findings by Andersson et al. [27], in which intrinsic motivation for changing personal substance use was reported to be limited in SUR prevention. That still notwithstanding, overwhelming evidence in the literature attests to the intrinsic motivation to quit as a protective factor against SUR [21]. Consistent with the findings of this study, intrinsic motivation for participants in a study by Chan et al. [28] was due to feelings of wanting to regain purpose in life.

Participants in this study reported the need for positive social interventions to prevent SUR. From this study’s findings, social interventions encompass the support obtained from families and friends. Participants who reported positive support from families and friends were satisfied with their recovery trajectory and involved their family members in managing areas of their lives that could trigger relapse, e.g., handling their money. Additionally, the results of this study also revealed the need to cut off peers who use substances and would serve as triggers of SUR. These findings are supported by Menon and Kandasamy [21], who reported positive social support, such as good family support, as contributing to long-term abstinence from substance use. According to the literature, a positive social support network predicts abstinence [21,29]. Conversely, negative social networks in the form of peer pressure from those involved in substance use and unsupportive families were associated with increased relapse to substance use [21].

The participants in this study also expressed the need for community interventions such as public education and public health campaigns to prevent substance use relapse. From the results of this study, public health education should be geared toward reducing stigma towards people with substance use problems. Participants in the current study decried discriminatory labelling utterances by the community as a cause of their relapses to substances, highlighting the necessity to address this issue. Additionally, participants in this study also suggested that public education and campaigns would raise the needed community awareness about the dangers associated with substance use, thus prompting actions to curtail the availability of drugs in the community, which in turn would help to prevent substance use and SUR. In line with the findings of this study, Saloner et al. [30] highlight the need for community education and sensitisation campaigns to raise awareness of the dangers of substance use. Additionally, participants in the current study suggested that such campaigns would reduce the availability of substances in the community, which studies report to be contributing to SUR [31,32]. Some studies have also reported public education as an important vehicle to use in combating stigma towards people with SUD, a strategy credited for the reduction of the same in HIV/AIDS and supported by the results of the current study [33].

Furthermore, this study suggests some of the healthcare interventions that can be used to prevent substance use relapse. The participants revealed the need to have specialised rehabilitation facilities for SUDs. They decried the double impact they endure from the stigma of using substances and that of receiving their rehabilitation services from a psychiatric hospital. As reported in some studies, the current study discovered that the public stigmatises psychiatric hospitals and labels those who access their services as “mad” [34]. This stigma hampers those with SUDs from seeking assistance in psychiatric hospitals, which contributes to SUR [34,35]. To prevent this, the current study’s participants suggested the need for independent specialised rehabilitation facilities for SUDs, something which is currently lacking in Botswana. Furthermore, participants in the current study reported that receiving their services from a psychiatric hospital limits their access to diverse and personalised SUD services. This assertion is supported by a study by Andraka-Christou et al. [36], in which participants emphasised the need for specialised SUD treatment centres. According to these authors, a specialised SUD centre would offer a variety of treatment options, personalised care, and adjunctive services beyond clinical treatment, such as employment assistance and financial literacy. Thus, a demand for SUD services that is disproportionately high relative to service provision is a cause of SUR, as participants in the current study echoed.

The participants in the current study expressed the need for activities such as occupational therapy, counselling, and psychoeducation to be extended beyond inpatient care and be offered on an outpatient basis to keep them engaged and prevent SUR. Literature supports such a programme, which it refers to as “day treatment”. When employing this approach, YSUs in the community are enrolled in an intensive outpatient programme and offered a variety of activities to prevent SUR [26,36]. The outpatient programme’s activities should be interesting and include, inter alia, occupational therapy and psychoeducation on addiction and the biology of the brain, which participants in the current study and other previous studies found interesting and helpful [37,38]. Other outpatient activities revealed in this study include continuing care through home visits, tracking YSUs post-discharge, and using technology, such as telephone calls, to monitor their progress. As per the findings of this study, the literature provides evidence that continuing care, e.g., tracking and assessment and home visits, is an essential component of post-treatment checkups to monitor recovery progress and prevent SUR [35,39,40,41,42]. Gonzales et al. [43] further added the significance of using technology such as the mobile texting intervention to buffer the tendency toward relapse as part of continuing care.

### Relevance of the Study to Clinical Practice and Research

While substance use is common in the youth demographic, the issue of relapse following periods of abstinence raises considerable concern, and prevention efforts are often neglected in various countries, including Botswana. This study presents findings on the perceptions that youth substance users have regarding substance use relapse prevention for individuals with substance use issues. The current research offers valuable insights into SUR prevention that can be used to develop evidence-based interventions, such as SUR prevention programmes. By exploring perceptions of SUR and how it can be prevented among the youth in Lobatse, this research can facilitate the creation of more tailored recovery and prevention strategies. Mental health professionals can leverage the findings of this study to design customised interventions that cater for the YSUs’ unique circumstances, ultimately increasing the likelihood of sustained recovery and relapse prevention.

In terms of research, available literature on mental health, including substance use problems in Botswana, provides inadequate guidance to inform policy and practice. Thus, the findings of this study close the knowledge gap that exists on the experiences of SUR among youth with substance use problems in Lobatse, Botswana. Furthermore, the study adds to the body of knowledge and can be beneficial in stimulating further research on the topic, which is necessary for continuing advancements in mental health.

## 5. Conclusions

This study’s findings present distinct causes of SUR that perpetuate and challenge recovery efforts for the youth wishing to abstain from substances. Together, they illustrate how diverse and multifaceted SUR is, and they call for appropriate, targeted SUR prevention strategies, as this study suggests. Strategies to prevent SUR include individual interventions, such as engaging in interesting activities to distract from thoughts about drugs, and social interventions, such as family and public education, to create a conducive environment that supports YSU’s recovery trajectory. Additionally, rigorous healthcare interventions such as tracking YSUs post-discharge, home visits, and the provision of intensive outpatient relapse prevention activities are needed if the battle against SUR is to be won.

### Strengths and Limitations of the Study

To the best of the researchers’ knowledge, this is the first study to be conducted in Botswana to explore and describe the perceptions of the youth on the prevention of substance use relapse in this age group. The study’s findings provide valuable insights into youth perceptions of SUR and can be extrapolated to similar contexts. The findings can be used to develop SUR prevention interventions, such as programmes, guidelines, and policies. However, the study gives context-specific insights and cannot be generalised to all YSUs in different contexts. Purposive sampling uses the researcher’s judgement to select participants who are viewed as better suited to address the aims and objectives of the study, introducing bias and resulting in skewed outcomes that might not accurately represent the views or behaviours of the broader substance-using population. Additionally, participants might have given socially desirable responses regarding their use of substances and relapses, due to fear of stigma, thus negatively impacting the results. Since the study was conducted at a referral institution, some participants might have been from other regions of the country, and the results may not be completely representative of the community dwelling in Lobatse.

## Figures and Tables

**Table 1 ijerph-23-00062-t001:** Demographic profile of the participants.

Participant	Age	Sex	Level of Education	Diagnosis
1	23	Male	Secondary	SUD
2	24	Male	Secondary	SUD
3	23	Male	Secondary	SUD, Anxiety Disorder
4	20	Male	Tertiary	SUD
5	21	Male	Primary	SUD
6	23	Female	Tertiary	SUD
7	24	Male	Tertiary	SUD
8	23	Male	Secondary	SUD
9	18	Male	Secondary	SUD
10	24	Male	Tertiary	SUD
11	20	Female	Tertiary	SUD
12	24	Male	Secondary	SUD
13	24	Female	Tertiary	SUD
14	18	Female	Secondary	SUD
15	22	Female	Tertiary	SUD, Depression

**Table 2 ijerph-23-00062-t002:** Themes and subthemes.

Themes	Subthemes
1. Perceptions of youth substance users on the causes of substance use relapse	1.1 Psychological causes of substance use relapse
	1.2 Social causes of substance use relapse
	1.3 Healthcare barriers to prevent substance use relapse
2. Perceptions of youth substance users on the prevention of substance use relapse	2.1 Individual interventions to prevent substance use relapse
	2.2 Social interventions to prevent substance use relapse (social support)
	2.3 Community interventions to prevent substance use relapse (community awareness and education)
	2.4 Healthcare interventions to prevent substance use relapse

## Data Availability

The data to support the findings of this study are available on request from the corresponding author (W.G.) on reasonable request.

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
