# Peer review of "Perceptions of Youth Substance Users on Substance Use Relapse Prevention: A Qualitative Study in Lobatse, Botswana"

_ijerph, 2025, doi:10.3390/ijerph23010062_

Round 1
Reviewer 1 Report
Comments and Suggestions for Authors
Line 40-41 can be removed
In introduction authors are mandated to add data such as prevalence of substance use, SUR
In methods section what authors mean by contextual research
In line 70 how relapse was defined among those patients
In line 82, the age groupe was between 18-24, can you precise why it was up to 24
Please see the additional comments in the attachment.

Author Response
|
Response to Reviewer Comments
Title of the study: Perceptions of youth substance users on substance use relapse prevention: A qualitative study in Lobatse, Botswana Wada Gaolaolwe 1,4,*, Miriam M. Moagi 2, Gaotswake Patience Kovane 1 and Leepile Sehularo3
|
||
|
1. Summary |
|
|
|
Thank you very much for taking the time to review this manuscript. Please find the detailed responses below and the corresponding revisions/corrections highlighted/in track changes in the re-submitted files. |
||
|
|
|
|
|
|
|
|
|
Point-by-point response to Comments and Suggestions for Authors
Reviewer 1 |
||
|
Comments 1: [Line 40-41 can be removed] |
||
|
Response 1: Thank you for pointing this out. I/We agree with this comment. Therefore, I/we have removed the sentence which was the last on page 1, under the introduction.
|
||
|
Comments 2: In introduction authors are mandated to add data such as prevalence of substance use, SUR |
||
|
Response 2: Agree. I/We have, accordingly, done as advised. The prevalence on substance use and SUR among youth has been added. The changes are in the introduction line 37 to 59 (page 1-2).
Comment 3: In methods section what authors mean by contextual research Response 3: The word, “contextual” has been removed to read, “qualitative, exploratory and descriptive research design”, because “contextual” is already implied within the other terms and need no emphasis. The corrections are in page 3, line 100 under methods (research design).
Comment 4: In line 70 how relapse was defined among those patients Response 4: The patients experiencing relapse have been defined in line 108-110, page 3 under study setting. Comment 5: In line 82, the age group was between 18-24, can you precise why it was up to 24 Response 5: An explanation on this category has been made, i.e., that it is in line with the definition of the United Nations (UN), which places the youth category between 18 and 24 years. Line 128-129, page 3. Comment 6: Please see the additional comments in the attachment. Response: The additional comments in the attachment have been addressed as follows (from comment 7:
Comment 7: The authors should provide more details on inclusion/exclusion criteria, recruitment process, justification of sample size (e.g., saturation), and participant demographics. Response: Inclusion and exclusion criteria have been added, including the recruitment process and justification for sample size. The information can be found on pages 3-4, lines 127-140. NB: Participants’ demographics are already there in the manuscript and can be found on page 6, line 250. Comment 8: It is unclear how interviews were conducted (duration, setting, language, recording tools, ethical approvals). Response 8: The issues raised have all been addressed. Duration of interviews was already there and is under data collection method, line 173-174, page 4. The setting is under the heading “study setting”, line 104, page 3. An audio recorder was used- line 170, page 4. Information on ethical considerations showing ethical approvals has been added- line 180-187, page 4-5. Comment 9: Thematic analysis needs more transparency (the authors should describe how themes were generated, coding procedures, number of coders, inter-rater reliability, and how discrepancies were resolved). Response: Detail has been provided by providing information on the phases of thematic analysis, with information under these steps addressing the issues raised- Line 198-223, page 5. NB: For inter-rater reliability in qualitative studies, the researcher and co-coder had a consensus meeting to compare their identified themes to evaluate agreement and ensure that the themes are robust and consistently interpreted- Line 233-234, page 5 under trustworthiness. Comment 10: The background section briefly mentions the issue in Botswana but lacks deeper context (national prevalence, youth-specific data, current interventions, policy framework). Response: More information has been added as advised for depth and is in line 44-59, page 2. |
||
|
Comment 11: The discussion could more strongly link findings to existing literature, especially African or regional studies on relapse. Response: Information on the Western countries, Africa (sub-Saharan Africa) and Botswana have been included. Lines 37-45, pages 1-2.
Comment: There is no clear indication of a guiding theoretical framework (e.g., relapse prevention theory, social learning theory). A theoretical background would strengthen interpretation of findings. Response: As the study is qualitative research, it is highly inductive, emphasizing understanding of a phenomenon (SUR) by exploring the varied, lived experience of youth substance users. Thus, the researchers were cautious about using theory in the qualitative research process because of its potential to impose meaning and subsequently alter the organic understanding of the phenomenon being investigated, a stance that is also supported by literature. So, the insights shared in this qualitative study are typically grounded in the experiences and perspectives of participants, supported by references to existing literature. |
||
|
|
||
Reviewer 2 Report
Comments and Suggestions for Authors
The goal of this manuscript was to summarize descriptive data from a qualitative cross-sectional study of young substance users’ perceptions of (a) the multiple causes of substance use relapse and (b) prevention strategies for substance use relapse. The manuscript summarized themes and sub-themes emerging from the semi-structured interviews of 15 participants. The topic of the manuscript addresses an important public health issue, and the study has multiple strengths. However, attention to several issues would strengthen further the manuscript. Specifically:
- The manuscript is generally well-written and it addresses important public health and clinical issues pertinent to young adults with a history of unhealthy substance use patterns. The study is a useful extension of the available research and practice literature examining a question in a specific vulnerable population. The study is methodologically rigorous and the study’s findings are discussed with a thorough integration of existing research and practice-related literature.
- The rationale for the study could be strengthened further by expanding the introduction to include the integration of a theoretical perspective or conceptual model regarding substance use relapse that is guiding the study. In addition, the introduction could be strengthened by a clear statement describing the gap in the available research or practice literature that the present study is attempting to address.
- It is not surprising that the two major themes generated in the qualitative analysis of the semi-structured interview content were the ones generated. After all, the questions that guided the semi-structured interview (see section 2.4) specifically referred to these themes. This point reduces the “exploratory” nature of the study.
- The study appears to use purposive sampling from a single inpatient psychiatric hospital in Lobatse, Botswana. The authors should comment on ways in which the sampling strategy may have impacted participants’ responses to the semi-structured interview and the themes generated. Would the responses have varied substantially if a community-dwelling sample had been included in the study?
- The discussion section of the manuscript could also be strengthened further. Specifically, the discussion section would benefit from a clear statement regarding how the present manuscript adds meaningfully to the existing research and practice literature related to young adults’ experience or perception of substance use relapse. In addition, the results should be interpreted in the context of existing conceptual models of relapse of substance use. Finally, given the cultural context of the data collection site and the lives of the participants, the results of the study should be discussed further with reference to that cultural and developmental context.
Author Response
Reviewer 2
Comment 1: The introduction could be strengthened by a clear statement describing the gap in the available research or practice literature that the present study is attempting to address.
Response 1: Information has been included in the introduction, indicating the gap in research that led to conducting the study. Line 49-59, page 2.
Comment 2: It is not surprising that the two major themes generated in the qualitative analysis of the semi-structured interview content were the ones generated. After all, the questions that guided the semi-structured interview (see section 2.4) specifically referred to these themes. This point reduces the “exploratory” nature of the study.
Response 2: Though the results have been coalesced into two themes, each theme has subthemes. On the exploratory part, the questions were open-ended, prompting participants to elaborate on their thoughts and experiences. This encouraged deeper exploration, not limiting responses to the main questions asked, allowing the exploratory nature of the study. Furthermore, the researcher indicated that communication skills were used such as clarification, exploration, etc., to get the depth needed. Line 174-176, page 4
Comment 3: The study appears to use purposive sampling from a single inpatient psychiatric hospital in Lobatse, Botswana. The authors should comment on ways in which the sampling strategy may have impacted participants’ responses to the semi-structured interview and the themes generated. Would the responses have varied substantially if a community-dwelling sample had been included in the study?
Response 3: The issues raised have been added and are addressed in the limitations of the study. Line 522-530, page 12.
Comment 4: The discussion section of the manuscript could also be strengthened further. Specifically, the discussion section would benefit from a clear statement regarding how the present manuscript adds meaningfully to the existing research and practice literature related to young adults’ experience or perception of substance use relapse. In addition, the results should be interpreted in the context of existing conceptual models of relapse of substance use. Finally, given the cultural context of the data collection site and the lives of the participants, the results of the study should be discussed further with reference to that cultural and developmental context.
Response: Additions has been made after the discussion to include a heading on the relevance of the study to practice and research. Lines 484-502, page 11-12. The cultural context of the study has been provided in line with how the community stigmatizes the data collection site, leading to reluctance to seek help and consequently relapses. Line 448-455, page 11.
As the study is qualitative research, it is highly inductive, emphasizing understanding of a phenomenon (SUR) by exploring the varied, lived experience of youth substance users. Thus, the researchers were cautious about using theory in the qualitative research process because of its potential to impose meaning and subsequently alter the organic understanding of the phenomenon being investigated, a stance that is also supported by literature. So, the insights shared in in this qualitative study is typically grounded in the experiences and perspectives of participants, supported by references to existing literature
Reviewer 3 Report
Comments and Suggestions for Authors
The article aims to reveal how former substance users perceive the reasons for relapse and how they perceive possible relapse prevention methods.
The authors conducted qualitative interviews with 14 outpatient department patients in Steswana who had previously attempted to quit substance use but experienced a relapse.
While the research aim is important, it is not extremely novel from an international viewpoint. It has been discussed by other scholars, and the authors refer to many of these studies as well. Reading the conclusion chapter, I realized that most of the facilities and circumstances that would help prevent relapse are unavailable in Botswana. Therefore, the study's results, which highlight the importance of different facilities in decreasing the probability of relapse, may be significant for Botswana's health and social care systems and for other countries with similar systems.
In the introduction chapter, I would like to see a broader review of the literature discussing the reasons for relapse. I also suggest providing a brief overview of the health and social services available to drug users in Botswana. This would better highlight the importance of the study.
The methodology is relevant and correctly presented. In the methodology chapter (or in the findings chapter), I missed a description of the substance-using carrier of respondents, as the length of substance use or the type of substance used might impact the likelihood of quitting.
The findings chapter presents results by different subtopics. Did these subtopics simply emerge from the interviews, or was there any hypothesis or scientific consideration behind analysing results by these dimensions? I miss some explanation here.
When reading the reasons for relapse, they are very similar to the reasons for becoming a substance user. Are there any differences in the reasons for starting substance use or for relapse, or are the two types of reasons related? Some explanatory sentences would be useful.
In some cases, such as subtheme 2.4, it would be better to summarize the main points of the quotes in your own words rather than only presenting the main statement in the quotes.
The discussion chapter presents and discusses the main results and conclusions of the study well.
While the reasons for relapse are not extremely novel, the study reveals significant gaps in care provision. The article is interesting and well written, but a few additions would improve its quality.
Author Response
Reviewer 3
Comment 1: In the introduction chapter, I would like to see a broader review of the literature discussing the reasons for relapse. I also suggest providing a brief overview of the health and social services available to drug users in Botswana. This would better highlight the importance of the study.
Response: The reasons for relapse have been added to the introduction, including the services for substance users in Botswana. Lines 76-93. Page 2-3. (NB: Also, a more broader overview of substance use relapse is articulated comprehensively integrated in the discussion).
Comment 2: The methodology is relevant and correctly presented. In the methodology chapter (or in the findings chapter), I missed a description of the substance-using carrier of respondents, as the length of substance use or the type of substance used might impact the likelihood of quitting.
Response 2: The demographics for the participants are covered under the results. However, they only included age, sex, level of education, and diagnosis. Line 249, page 6.
Comment 3: The findings chapter presents results by different subtopics. Did these subtopics simply emerge from the interviews, or was there any hypothesis or scientific consideration behind analysing results by these dimensions? I miss some explanation here.
Response: The findings were inductive (emerging from the data) and there was no hypothesis as this was a qualitative study. The discussion is according to themes, which provided a framework for discussing the findings, allowing researchers to connect their results to existing literature.
Comment 4: When reading the reasons for relapse, they are very similar to the reasons for becoming a substance user. Are there any differences in the reasons for starting substance use or for relapse, or are the two types of reasons related? Some explanatory sentences would be useful.
Response 4: Though reasons are generally similar, the explanation in lines 64-67 addresses the concern. The reason for relapse may not just be, e.g., to cope with stress, as it can be with just a substance use, but may be because of addiction, which gives a stronger physiological reason for use. More information on the causes of relapse are also captured in 76-93.
Comment 5: In some cases, such as subtheme 2.4, it would be better to summarize the main points of the quotes in your own words rather than only presenting the main statement in the quotes.
Response 5: The quotes are presented here as the results. There have been discussed in the researchers’ own words under discussion. From line 367, page 9.
Comment 6: The discussion chapter presents and discusses the main results and conclusions of the study well.
Comment 7: While the reasons for relapse are not extremely novel, the study reveals significant gaps in care provision. The article is interesting and well written, but a few additions would improve its quality.
Response: The reviewers' comments have been incorporated to improve the quality.